# Diabetic Plantar Foot Segmentation in Active Thermography Using a Two-Stage Adaptive Gamma Transform and a Deep Neural Network

**DOI:** 10.3390/s23208511

**Published:** 2023-10-17

**Authors:** Zhenjie Cao, Zhi Zeng, Jinfang Xie, Hao Zhai, Ying Yin, Yue Ma, Yibin Tian

**Affiliations:** 1College of Mechatronics and Control Engineering, Shenzhen University, Shenzhen 518000, China; 2020110516010@stu.cqnu.edu.cn (Z.C.); yue.ma@ncepu.edu.cn (Y.M.); 2College of Computer and Information Science, Chongqing Normal University, Chongqing 401331, China; 2021110516026@stu.cqnu.edu.cn (J.X.); zhaihao@cqnu.edu.cn (H.Z.); 3Shunde Hospital, Southern Medical University, Foshan 528000, China; 4Department of Rehabilitation Medicine, The Second Affiliated Hospital of Chongqing Medical University, Chongqing 400010, China; 300735@cqmu.edu.cn

**Keywords:** diabetic foot, active thermography, image segmentation, adaptive gamma transform, deep neural network

## Abstract

Pathological conditions in diabetic feet cause surface temperature variations, which can be captured quantitatively using infrared thermography. Thermal images captured during recovery of diabetic feet after active cooling may reveal richer information than those from passive thermography, but diseased foot regions may exhibit very small temperature differences compared with the surrounding area, complicating plantar foot segmentation in such cold-stressed active thermography. In this study, we investigate new plantar foot segmentation methods for thermal images obtained via cold-stressed active thermography without the complementary information from color or depth channels. To better deal with the temporal variations in thermal image contrast when planar feet are recovering from cold immersion, we propose an image pre-processing method using a two-stage adaptive gamma transform to alleviate the impact of such contrast variations. To improve upon existing deep neural networks for segmenting planar feet from cold-stressed infrared thermograms, a new deep neural network, the Plantar Foot Segmentation Network (PFSNet), is proposed to better extract foot contours. It combines the fundamental U-shaped network structure, a multi-scale feature extraction module, and a convolutional block attention module with a feature fusion network. The PFSNet, in combination with the two-stage adaptive gamma transform, outperforms multiple existing deep neural networks in plantar foot segmentation for single-channel infrared images from cold-stressed infrared thermography, achieving an accuracy of 97.3% and 95.4% as measured by Intersection over Union (IOU) and Dice Similarity Coefficient (DSC) respectively.

## 1. Introduction

Diabetes, a chronic endocrine disorder, is characterized by insufficient insulin production, thereby leading to elevated glucose concentrations in the blood, causing potential harm to blood vessels and nerves [1]. In 2021, the global adult diabetic population was approximately 537 million, an increase of nearly 50 million from 2019 [2]. Diabetes can cause damage to various body parts, with diabetic foot disease being a prevalent complication [1,3].

In clinical practice, plantar foot assessment is important for early diagnosis, progression monitoring, and devising treatment plans. This assessment encompasses a thorough evaluation of a patient’s medical history and may be supplemented with additional investigations. A common manifestation in patients with diabetic feet is swelling, which can vary in symptoms, for example, itchiness and pain, depending on the individual case. The affected plantar foot regions typically exhibit irregular structures and undefined outer boundaries [4]. The lifetime risk of diabetic patients developing foot ulcers is around 12–25% [5]. The visual appearance of this diseased area and the surrounding skin is dependent on different disease stages, potentially exhibiting signs like redness, crust formation, blistering, granulation, mire, hemorrhages, and scaly skin [3]. Substantial research has been carried out to utilize color images and machine learning for diabetic foot ulcer detection and classification [6,7]. In particular, Yap et al. provided a large public dataset (DFUC2020) and systematically compared various deep learning models for diabetic foot ulcer detection in color images and showed that the Cascade Attention Network (CA-DetNet), Faster R-CNN with deformable convolution, and ensemble method were among the best performing deep learning methods for the DFUC2020 dataset [6]. However, the early stages of plantar foot symptoms are often inconspicuous in color images, leading to diagnoses often being given in the intermediate or late stages, posing a considerable health risk. Studies have shown that it is important to screen for diabetes to reduce the lead time between diabetes onset and clinical diagnosis and to allow for prompt treatment [8].

Temperature variations in the plantar region have been shown to be related to diabetic foot problems [9]. Various techniques have been studied for plantar temperature measurements, such as infrared thermometry [10,11], liquid crystal thermography [12], and, more recently, infrared thermography [13,14,15]. Infrared thermometry is a low-cost and widely accessible technique for temperature measurements, and can be used at a patient’s home, but it only measures point by point. When many points need to be monitored, the procedure becomes complicated and unreliable [10]. Liquid crystal thermography is another low-cost technique that can provide rich color representations of entire plantar temperature distributions, but it only works when the plantar foot is in good contact with the thermochromic liquid crystals, and its response is slow [12]. Due to the limitations of these two techniques, the use of infrared thermography for non-contact, high-resolution, and rapid temperature measurements of diabetic feet has increased significantly in the past 10 years [13,15,16]. It should be noted that another diagnostic technique, Doppler ultrasonography, has been commercially available for monitoring peripheral vascular disease and other artery abnormalities that often occur in diabetic feet [16]. Doppler ultrasonography provides information on blood flow issues in peripheral arteries and may be more accurate for diabetic foot diagnosis, but it requires a gel between the ultrasonic transducers and patient skin. Compared to infrared thermography, it is less comfortable for patients and more difficult to operate for care givers. In Denmark, it has been estimated that integrating infrared thermography into diabetic foot care can potentially save approximately 20% of the total annual cost in the long run, with increased costs in the short term [17]. The exact numbers may vary, but the overall pattern is expected to be the same in other countries with similar health care systems.

Plantar thermogram acquisition requires patients to remove their shoes and socks while sitting or in the supine position for a certain period (for example, 10 to 15 min). There are two procedure types utilizing infrared thermography in diabetic foot research. One uses passive infrared thermography, which takes one or a sequence of thermal images when the patient’s feet are in a resting condition to reach thermal balance, as described above, without external stimulation. The other utilizes one special type of active infrared thermography, in which thermal or pressure stress is applied to the patient’s feet before a long sequence of thermal images are acquired while the feet recover from the stress to reach thermal balance [13,18]. The former approach mainly analyzes the plantar region temperature of each foot, the temperature asymmetry between each patent’s two feet, or the temperature distribution in each foot without disturbance, while the latter approach mainly focuses on the dynamic temperature changes after applying thermal or pressure stress [13]. Obviously, passive infrared thermography is more comfortable for patients and easier to carry out, but active infrared thermography provides much richer temporal information and may better reveal early diabetic foot complications at the cost of a more complex procedure and lower patient comfort.

Accurate segmentation of plantar feet is crucial for temperature analysis in infrared thermography. Threshold segmentation methods yield satisfactory results when there is substantial contrast between the background and the plantar region, as is often the case in well-controlled passive infrared thermography. Kaabouch et al. experimented with five automated threshold methods for segmenting the biped from the background, namely, histogram shape, clustering, entropy, object attributes, and a genetic algorithm [19]. Liu et al. showed that an unbounded active contour model still requires manual adjustments for poor image contrast or unobstructed body parts like ankles [20]. Bougrine et al. proposed a method that employs an improved active contour inclusive of a priori shape information and plantar contour mapping for segmenting plantar infrared thermograms, which outperformed traditional methods like the classical snake, region growth, the active contour method, level set, and graph cut [21,22]. However, it underperforms when cold regions are present in the heat map, and its parameter settings require optimization. Feature selection and optimization combined with traditional machine learning methods, such as support vector machines and AdaBoost, have also been explored for plantar foot detection in thermal images [23].

As traditional methods necessitate human intervention and relevant knowledge for adjustments, deep learning methods have been investigated for plantar foot detection or segmentation in thermal images [23,24,25,26]. Some of these studies employed existing deep neural networks or their variations, such as the Full Convolutional Network (FCN) [27], DenseNet [23], MobileNet [23], ResNet [23], VGG [23], SegNet [28], and UNet [29], and others suggested more sophisticated networks, such as Double Encoder-ResUNet (DE-ResUNet) [24], to achieve impressive segmentation results. However, the datasets in these studies were acquired via passive infrared thermography to produce good image contrast between the background and plantar feet. Additionally, some of these studies used color or depth images as separate input channels in addition to thermal images to help improve segmentation accuracy [24,25]. Adding additional color or a depth camera in the image acquisition step requires complex inter-modal multi-camera calibration apparatus and procedures for precise color–thermal or depth–thermal image alignments. In addition, the parameters of such multi-camera setups are prone to being affected by environmental factors such as mechanical shocks, which complicates their maintenance for long-term usage. In addition, color camera usage is more likely to lead to privacy concerns for some patients compared to the usage of a thermal camera alone.

In this study, we investigate plantar foot segmentation methods for thermal images obtained via cold-stressed active infrared thermography without the complementary information from color or depth channels. To better manage the temporal variations in image contrast when planar feet are recovering from cold immersion, we introduce an image pre-processing method using a two-stage adaptive gamma transform to alleviate the impact of such contrast variations. To improve upon existing deep neural networks for segmenting planar feet from cold-stressed infrared thermograms, a new deep neural network, the Plantar Foot Segmentation Network (PFSNet), is proposed. It combines the fundamental U-shaped network structure, a multi-scale feature extraction module, and a Convolutional Block Attention Module (CBAM) with a feature fusion network. The PFSNet, in combination with the two-stage adaptive gamma transform, outperforms multiple existing deep neural networks in segmentation accuracy for single-channel infrared images from cold-stressed active infrared thermography.

## 2. Methods

A thermal camera with a high resolution and a high sensitivity was custom-built to acquire a sequence of plantar foot thermograms from volunteer subjects after their feet were subjected to cold stress while they rested in a supine position. The thermogram sequence captured the plantar foot temperature variations during the thermal recovery process (lasting over 10 min) and exhibited considerable contrast variations over time. A two-stage adaptive gamma transform was utilized to handle the temporal contrast variations in the thermal images. The transformed images were fed into a new deep neural network that consists of a classic U-shaped network structure with a multi-scale feature extraction module and a feature fusion module. These steps are explained in detail in the following sections.

### 2.1. Thermography Setup

The typical human skin temperature ranges from 36 °C to 37.5 °C, corresponding to a peak wavelength of 9.5μm for the related heat effusivity [30]. A custom-built un-cooled vanadium oxide infrared camera from NovelTEQ (Nanjing, China) (Figure 1), with working wavelengths of 8–14μm, was utilized. The camera has 640 × 512 pixels, a Noise Equivalent Temperature Difference (NETD) of less than 40 mK (i.e., a temperature sensitivity < 0.04 °C), an ADC bit depth of 16, a maximum frame rate of 50 Hz, and electrically controlled focusing. The infrared camera resolution is higher than that used in many recent infrared thermography studies of diabetic feet, which employed commercial cameras such as an FLIR E60 with 320 × 240 pixels or an FLIR One Pro with 160 × 120 pixels, and its thermal sensitivity of less than 40 mK is better than the 50 mK of the FLIR E60 and the 70 mK of the FLIR One Pro [18,22,24]. To acquire accurate temperature values and facilitate quantitative comparison among differing experimental results, two black bodies were placed in the camera field of view for calibration, with temperatures preset at two distinct values of 28 °C and 36 °C (as shown on the two sides of the image in Figure 1). The acquired thermogram data were first converted into temperature data using the pixel values of the two black bodies, and then transformed into grayscale images (see Section 2.4 for details).

It should be noted that a high-end cooled thermal camera can offer better temperature sensitivity and measurement precision than the NovelTEQ camera employed here, but at a significantly higher cost. Meanwhile, the NovelTEQ camera, in combination with the two calibration black bodies, provides better image resolution, temperature sensitivity, and measurement precision than the thermography hardware utilized in several previous diabetic foot studies [18,22,24].

### 2.2. Experimental Procedure

Both passive and active infrared thermography have been employed for diabetic foot detection. We focus on active thermography due to its potential to provide richer data compared to its passive counterpart. The experimental conditions were carefully controlled, with the ambient temperature of the room maintained at approximately 22 °C. The plantar feet were cooled to a temperature below room temperature through two different methods: direct cooling via cold water and gel pads and encasing feet in a plastic bag before subsequent cooling with cold water. Each method was subjected to trials at various temperatures and with various cooling durations.

Subjects were positioned in a relaxed resting supine position with a patient-supporting bed-like setup similar to a recent study [18], with their ankles enveloped in black foam to isolate the rest of the body from the area of interest. The infrared camera was positioned at roughly 0.5 m from the patient’s feet, facilitating the capture of the plantar foot temperature recovery process over a span of approximately 10–15 min at a rate of one image per second (1 Hz). It is essential to allow time for the plantar feet to cool down and to record the entire temperature recovery process. A total of 600 to 900 thermograms of both feet in the middle of the camera field of view were taken for each subject.

### 2.3. Experimental Subjects

Healthy volunteers were recruited from staff members of the Shunde Hospital of Southern Medical University, and diabetic volunteers were recruited from patients at the endocrinology clinic of the same hospital with flyers outlining the study. Volunteers showing signs of foot deformation or skin damage were not allowed to enroll. Ten diabetic volunteers (six males and four females, average age: 45.3 ± 3.4 years) and twelve healthy ones (six males and six females, average age: 44.5 ± 4.8 years) participated in and finished the study. The number of volunteers was small. Since one thermogram sequence was obtained for each subject, the total number of raw images acquired exceeded 6000 and 7200 for the healthy and diabetic subjects, respectively.

The study adhered to the Declaration of Helsinki and was approved by the Medical Ethics Committee of Southern Medical University in advance. Signed consent forms were obtained from all volunteers after detailed explanations of the purpose, procedures, potential discomfort, and expected outcomes.

### 2.4. Plantar Foot Thermal Images

The fixed temperature window methodology was adopted to convert radiation values into grayscale values [31]. Given the relatively limited temperature range in the experimental context, a temperature window value of 10 °C was selected for the conversion of radiation values to grayscale values.

Throughout the experiment, the background temperature could be considered as constant, while the plantar temperature progressively increased from a point lower than the background temperature to body temperature. This temperature evolution gave rise to two characteristic scenarios within our image dataset. In the later stages of the experiment, the plantar temperature significantly surpassed the background temperature, resulting in a strong image contrast, as in Figure 2a. The complete foot edge could be obtained using conventional segmentation methods. In the earlier stages of the experiment, the plantar temperature was either lower than or comparable to the background temperature, leading to a poor image contrast and the entire image appearing predominantly dark, as shown in Figure 2b. Traditional methods failed to extract the complete foot edges in this case.

As the above temporal evolution is relatively slow, the difference between consecutive thermograms in each image sequence is relatively small. For thermogram segmentation investigation purposes, only one thermogram from every 10 frames was used. Blurry images and images without proper calibration targets or foot posture were also excluded.

### 2.5. Two-Stage Adaptive Gamma Transform

A two-stage adaptive gamma transform was proposed for image pre-processing to reduce the impact of temporal contrast variations in thermal images due to cold stress and the subsequent recovery described above. It adaptively calculates the appropriate gamma factor (γ) based on image characteristics, thus controlling the transform’s scaling degree and achieving optimal compensation effects. The γ factor is calculated based on the cumulative distribution function (CDF) of the relative value of the thermal image.

A typical CDF curve of a low-contrast dark image is shown in Figure 3, in which *x*_0_ corresponds to the median of the CDF (i.e., 0.5). The relative ratio of temperature difference *β* is defined as:(1)$$\beta =\frac{x_{1}-x_{0}}{x_{0}}$$
where *x*_1_ is the pixel value corresponding to a CDF of 0.75. Gamma (γ) is computed as
(2)$$\gamma =\left\{ \begin{array}{l} 0.5+x_{0},x_{0}\leq 0.5\\ \frac{1}{1.5-x_{0}},x_{0}>0.5 \end{array}\right.\;\mathrm{for}\;\beta \leq 1$$
(3)$$\gamma =\left\{ \begin{array}{l} \frac{1}{0.5+x_{0}},x_{0}\leq 0.5\\ 1.5-x_{0},x_{0}>0.5 \end{array}\right.\;\mathrm{for}\;\beta \;>1$$

The classical gamma transform is
(4)$$V_{out}={V_{in}}^{\gamma }$$
where *V_in_* is the original pixel value of the input image and *V_out_* is the transformed pixel value of the output image [32]. Each obtained experimental image sequence can be divided into early and late stages. In the early stages, the plantar temperature is lower than or close to the background temperature, the image is dark, and *β* is less than or equal to 1. In the late stages, the plantar temperature is higher than the background temperature, the image has higher contrast, and *β* is greater than 1. The proposed transform was implemented using different *γ* values for different *β* ranges.

### 2.6. Plantar Foot Segmentation Network (PFSNet)

To improve feature extraction from plantar foot thermograms, we propose a new deep neural network termed PFSNet. This design combines the fundamental U-shaped network structure, a multi-scale feature extraction module, and a CBAM with a feature fusion network. The U-shaped network is widely used as a backbone for various sophisticated deep neural networks for image segmentation [29,33,34]. As depicted in Figure 4, PFSNet can be partitioned into three core modules: the data input module, the feature extraction module, and the feature fusion module. These modules are explained in detail below.

#### 2.6.1. Input Module

Drawing inspiration from GoogLeNet [35], the PFSNet input module starts with a preliminary convolution (Figure 5). This stage is succeeded by the pooling and normalization layers. This reduces the dimensions of the initially large input images for subsequent layers.

#### 2.6.2. Feature Extraction Module

The feature extraction module consists of three main components: a five-tier encoder, a seven-tier decoder, and a series of nested dense convolutional blocks.

Given the restricted perceptual field of 1 × 1 or 3 × 3 filters, which fails to capture the global information, the output feature map of the preceding layer captures solely local features. The obvious approach is to expand the receptive field to gather more global data from the feature maps. The inflated convolution module (akin to an inception-like block) extends the perceptual field to extract local and non-local features via null convolution.

A novel RSU residual module was proposed by Qin et al. to capture in-stage multi-scale features in U^2^Net [34]. Figure 6 shows the RSU-L structure, where L is the number of encoder layers, C_in_ and C_out_ denote the input and output channels, and M denotes the number of channels in the internal layers of the RSU. This design change enables the network to extract features directly from multiple scales of each residual block. It is worth noting that the computational cost of the U-structure is low because most operations are applied to the down-sampled feature maps.
(1)Input layer. It transforms the input feature map, X(H × W × C_in_), into an intermediate map, F_1_(x), with C_out_ number of channels.(2)Using the intermediate feature map, F_1_(x), as input to extract and encode multi-scale contextual information U(F_1_(x). When L is larger, the RSU is deeper and has more pooling operations, with a larger perceptual area to extract richer local and global features. Configuring this parameter allows the extraction of multi-scale features from an input with an arbitrary spatial resolution. This process reduces the detail loss caused by direct up-sampling at large scales.(3)Fusion of local features and multi-scale features by summation F_1_(x) + U(F_1_(x)). *RSU7*, *RSU6*, *RSU5*, *RSU4*, and *RSU4F* are used in the encoder. The numbers 7, 6, 5, and 4, refer to the height *L* of the *RSU*. *L* is usually configured according to the spatial resolution of the input feature map. At *RSU4*, the feature map resolution is relatively low, and further down-sampling of these feature maps results in the loss of helpful context. Therefore, in the *RSU4* stage, *RSU4F* is used, where F denotes that the RSU-L is an extended version in which we replace the merging and up-sampling operations by increasing the number of inflated convolutions. This means that all intermediate feature maps of RSU-4F have the same resolution as the input feature maps. 

The decoder structure is similar to that of the encoder stage. Each decoder step takes a cascade of up-sampled feature maps from the previous step and from its symmetric encoder stage as inputs.

The nested dense convolutional block essentially enhances the semantic level of the encoder feature mapping. The skip path is described as follows: let RSU-L(*i*,*j*) denote the output of node RSU-L, where *i* is the down-sampling layer in the encoder and *j* is the convolutional layer of the dense block in the skip path. The feature mapping stack represented by RSU-L(*i*,*j*) is calculated as
(5)$$RSU-L(i,j)=\left\{ \begin{array}{l} H(RSU(i-1,j)),j=0\\ H([{\left[RSU(i,k)\right]}_{k=0}^{j-1},u(RSU(i+1,j-1))]),j>0 \end{array}\right.$$
where the function *H*(-) represents the convolution operation within the RSU-L residual module, followed by the activation function, the up-sampling layer *u*(-), and the connection layer [-]. Nodes at level *j* = 0 receive only one input from the previous encoder layer; nodes at level *j* = 1 receive two inputs, both from the encoder sub-networks; and nodes at level *j* > 1 receive j inputs, which are the outputs of the first *j* nodes in the same skip path. The last input is the up-sampled output from the lower skip path. All previous feature maps accumulate and reach the current node because we utilize dense convolutional blocks along each skip path.

To address the issue of sample scarcity and enhance small target detection efficiency, the Convolutional Block Attention Module (CBAM) is integrated into PFSNet. In the decoding stage, the CBAM is added when each RSU-L block is stitched for up-sampling to augment the model’s capacity to process the region’s features of interest and improve the segmentation effect.

#### 2.6.3. Deep Supervision by Multiple Side-Output Fusion

There are two challenges in the PFSNet framework: addressing the vanishing gradient issue to improve the convergence of deep networks and facilitating the learning of more insightful features from the lower to the higher levels. To tackle these issues, the Multi-Scale Optimization Fusion (MSOF) strategy is employed [36].
(6)$$\begin{array}{l} Y^{0,5}=Y^{0,1}⊕Y^{0,2}⊕Y^{0,3}⊕Y^{0,4}\\ Y^{5,0}=Y^{0,0}⊕Y^{1,0}⊕Y^{2,0}⊕Y^{3,0}⊕Y^{4,0}\\ Y^{5,5}=Y^{5,0}⊕Y^{0,5} \end{array}$$

As shown in Figure 7, we use the output nodes RSU7, RSU6, RSU5, RSU4, and RSU4F as fusion layer inputs. The feature map resolution is kept at the same size by expanding the convolution to obtain a multi-side output {*Y*^0,1^, *Y*^0,2^, *Y*^0,3^, *Y*^0,0^, *Y*^1,0^, *Y*^2,0^, *Y*^3,0^, *Y*^4,0^}. Afterwards, the results from both sides are superimposed to obtain the final feature map using Equation (6), where ⊕ denotes the concatenation operation. Again, RSU7 is followed by a sigmoid layer, and the fusion output *Y*^0,5^ can thus be generated. Therefore, five outputs are generated in the network, namely {*Y*^0,1^, *Y*^0,2^, *Y*^0,3^, *Y*^0,4^, *Y*^0,5^}, where *Y*^0,5^ is the fusion output of {*Y*^0,1^, *Y*^0,2^, *Y*^0,3^, *Y*^0,4^}. The same is applied to {*Y*^0,0^, *Y*^1,0^, *Y*^2,0^, *Y*^3,0^, *Y*^4,0^, *Y*^5,0^}, where *Y*^5,0^ is the fusion output of {*Y*^0,0^, *Y*^1,0^, *Y*^2,0^, *Y*^3,0^, *Y*^4,0^}, and {*Y*^5,0^, *Y*^0,5^, *Y*^5,5^}, where *Y*^5,5^ is the fusion output of {*Y*^5,0^, *Y*^0,5^}. Through the MSOF operations, multi-level feature information from all side-output layers is embedded in the final output *Y*^5,5^, which can capture finer spatial details.

### 2.7. Loss Function

The training loss function is defined as
(7)$$L=\sum_{m=1}^{M}w_{side}^{\left(m\right)}e_{side}^{\left(m\right)}+w_{fuse}^{}e_{fuse}^{}$$
where *e_side_* is the loss of the side output, *e_fuse_* is the loss of the final fusion output, and *w_side_* and *w_fuse_* are the weights of the two loss terms, respectively. For each term, we use the standard binary cross-entropy:(8)$$e=-\sum_{\left(r,c\right)}^{\left(H,W\right)}\left[P_{G(r,c)}\log(P_{S(r,c)})+(1-P_{G(r,c)})\log(1-P_{S(r,c)})\right]$$
where (*r*,*c*) are the pixel coordinates and (*H*,*W*) are the image height and width. *P_G_*_(*r*,*c*)_ and *P_S_*_(*r*,*c*)_ denote the pixel values of the ground truth and the predicted saliency probability map, respectively.

## 3. Experimental Results

### 3.1. Dataset

As two cooling techniques were employed in this study, this may have yielded inconsistent cooling effects, which consequently increase the image segmentation complexity. These irregular cooling effects may also reflect the temperature variations induced by pathological areas. Hence, experimental results incorporating both uniform and non-uniform cooling are included in this investigation. The dataset contains 1800 plantar foot thermal images. The original images were automatically cropped to exclude the black bodies. Data processing involved image flipping for data augmentation and applying the two-stage adaptive gamma transform. The dataset was partitioned into 85% for training, 5% for validation, and 10% for testing.

### 3.2. Implementation Details

PFSNet was implemented using Python 3.6 and PyTorch 1.7.0. The input image size and patch size were set to 512 × 512 and 4, respectively. Training was conducted for 100 epochs, with a batch size of 5, on an Nvidia V100 GPU with 16 GB memory. During the training phase, the batch size was 8 and the Stochastic Gradient Descent (SGD) optimizer was used, with a momentum of 0.9 and weight decay of 1 × 10^−4^ [37].

It took 52 ms for the trained PFSNet to segment one 512 × 512 thermal image, with an equivalent frame rate of approximately 20 fps.

### 3.3. Experimental Results of the Two-Stage Adaptive Gamma Transform

Examples of two-stage-adaptive-gamma-transformed images are shown in Figure 8. Common image quality metrics, e.g., information entropy, the peak signal-to-noise ratio (PSNR), and the mean squared error (MSE), were utilized to assess the image quality. These metrics were compared for images which were subjected to histogram equalization, the fixed gamma transform (constant γ), the adaptive gamma transform (γ computed via Equation (2)), and the two-stage adaptive gamma transform.

As shown in Table 1, the two-stage-adaptive-gamma-transformed images exhibit the lowest information entropy and the smallest MSE values (for information entropy and MSE, the smaller the value, the better the image quality), while their PSNR values are the second highest among all the compared methods (for PSNR, the higher the value, the better the image quality). The results indicate the proposed two-stage adaptive gamma transform is effective in improving the thermal image quality acquired via cold-stressed active infrared thermography. The two-stage adaptive gamma transform has much more impact on images where the plantar temperature is lower than or close to the background temperature (i.e., low contrast and dark images, as occur in the early stage of recovering from cold-stress, Figure 8a) than on those with high contrast (as occur in the late stage of recovering from cold-stress or in widely used passive thermography, Figure 8b).

### 3.4. Experimental Results of PFSNet

In order to quantitatively compare different deep neural network models, we employed Hausdorff Distance (HD) [38], Dice Similarity Coefficient (DSC) [39], and Intersection over Union (IOU) [40] as evaluation metrics to assess the segmentation results.

The effects of image flipping and the proposed two-stage adaptive gamma transform were first evaluated for PFSNet. In the first case, PFSNet was trained with neither image flipping nor the proposed transform; in the second one, PFSNet was trained with only the proposed transform; and in the last one, PFSNet was trained with both image flipping and the proposed transform. The trained models were evaluated with the same testing images. As shown in Table 2, all three segmentation accuracy metrics are significantly greater when both image flipping and the two-stage adaptive gamma transform are utilized or when the latter is utilized alone compared to those without any of them (one-tailed *t*-test, *p*-value < 0.01). Image flipping and the two-stage adaptive gamma transform pre-processing together improves the widely used IOU and DSC metrics by 1.7% and 1.6%, respectively. It should be noted that these numbers reflect the average improvement, while the two-stage adaptive gamma transform has a much more significant impact on a subset of the thermal images than the rest, as discussed above. For the subset of low-contrast thermal images, the improvement to segmentation accuracy by the proposed transform is certainly higher than the averaged values shown here. The advantage of utilizing the two-stage adaptive gamma transform is that no manual image separation or parameter turning is necessary as it automatically adapts to the statistics of input images.

The plantar foot thermogram dataset augmented by image flipping and pre-processed with the two-stage adaptive gamma transform was then evaluated using multiple deep neural network models, including UNet [29], UNet++ [33], U^2^Net [34], AttentionUNet [41], and TransUNet [42], with the outcomes shown visually in Figure 9 and quantitatively in Table 3. These UNet based models have been shown to achieve outstanding results in generic and medical image segmentation tasks. TransUNet incorporates the popular vision transformer into UNet to simultaneously extract global context information and enable precise localization [42]. The proposed PFSNet exhibits superior performance, with the highest accuracy of 96.2% (HD), 95.4% (DSC), and 97.3% (IOU), respectively. All three metrics for PFSNet are significantly greater than those from the other five compared deep learning models (one-tailed t-test, *p*-value < 0.01). Notably, PFSNet improves segmentation accuracy by 2.1% in DSC and 3.0% in IOU, respectively, over U^2^Net.

TransUNet also performs well in terms of the DSC and IOU metrics, with both metrics ranking the second highest among the six deep learning models. It seems that the attention mechanism is very useful for accurate segmentation of cold-stressed active thermograms, as the two best performing models both employ it. The 12 layers of the vision transformer in TransUNet incorporates the global self-attention mechanism into the encoder [43], while the proposed PFSNet combines CBAM and feature fusion at the decoder stage after multi-scale feature extraction. It should be noted that PFSNet is 14.8% smaller than TransUNet (parameter data size 168 MB vs. 197 MB) and takes 13.2% less time to train (327.9 ms vs. 377.8 ms for each epoch), while it still gains a segmentation accuracy boost of 0.5% in DSC and 2.2% in IOU, respectively.

## 4. Discussion

PFSNet was implemented in this study for binary classification tasks, specifically distinguishing between a plantar foot and its background. It can accurately detect foot edges in contrast-varying thermal images with efficiency, laying the foundation for further exploration of diabetic foot condition detection or classification. Future work may incorporate the two-stage adaptive gamma transform into the neural network as a pre-processing module to extend it to an end-to-end network, and expand its functionality to encompass a broader range of clinical investigations using active thermography.

### 4.1. Comparison to Other Recent Studies

Active infrared thermography has been widely used in non-destructive testing in industrial settings. A significant advantage is its capability to reveal temporally varying information of visually undetectable defects in the studied objects [44]. In diabetic foot infrared thermography research, thermally stressed active infrared thermography has been studied, including both cold and warm immersions of patient’s feet [13,45,46]. A recent study demonstrated the feasibility of using cold-stressed infrared thermography and the classic Support Vector Machine (SVM) machine learning method to detect early diabetic peripheral neuropathy [47]. Probably because of the more complex procedures, cold and warm immersion active thermography has not been thoroughly investigated for diabetic foot care. Most existing diabetic foot infrared thermography studies used passive thermal image acquisition protocols, sometimes in combination with other imaging modalities such as color or depth cameras to obtain multi-modal data [18,22,25].

For thermal-image-only plantar foot segmentation, Bougrine et al. obtained a DSC of 94% using a combination of the snake method and prior information [22], while the proposed method in this study achieved a DSC of 95.4%. Bouallal et al. achieved a 97% IOU using a sophisticated Double Encoder-ResUnet (DE-ResUnet) [24], close to the 97.3% IOU in this study, but they utilized both thermal and color images, while only thermal images were employed in this study. It should be noted that these metric values were based on different datasets and cannot be taken literally, but we believe the current study represents the state of the art in utilizing deep learning to segment thermal images from active infrared thermography without the assistance of other imaging modalities.

### 4.2. Limitations of the Current Study

There are several limitations to the current study, some of which will be addressed in future work.

First, the number of subjects at 22 was relatively small. We will continue to recruit more volunteers, with the goal of investigating the feasibility of using cold-stressed infrared thermography and deep learning for early diabetic foot detection. Accurate thermal image segmentation is an important step in this process. However, since more than 600 thermograms with temporal variations were collected for each subject, the number of images available for evaluating the proposed image segmentation method was not an issue.

Second, the procedures for cold immersion below room temperature and recovery for active infrared thermography are more complex than those for its passive counterpart for both the patients and care providers. In addition, some patients may experience mild discomfort due to the cold immersion. These factors can potentially hinder the adoption of the proposed procedure in clinical settings.

Third, passive infrared thermography can take one or a few snapshots of thermograms, while the proposed use of active infrared thermography involves acquiring a sequence of thermograms for a period of 10 min or more. Though active thermography provides much richer information than passive thermography, the amount of produced image data will be significantly larger than that for its passive counterpart if it is widely deployed, and image analyses require more sophisticated algorithms at higher computational cost.

At the current stage, we are mostly concerned with the technical feasibility of using cold-stressed infrared thermography and deep learning for early diabetic foot detection, and some of the ramifications of potential clinical adoption of the technology need to be addressed in future work.

## 5. Conclusions

A two-stage adaptive gamma transform was introduced as an image pre-processing method, tailored to the specific characteristics of temporal contrast variations in thermal images of plantar feet after cold immersion. A novel Plantar Foot Segmentation Network (PFSNet), employing a U-shaped network backbone, a multi-scale feature extraction module, and a Convolutional Block Attention Module (CBAM) with a feature fusion network, was proposed to better extract foot contours in thermal images from active thermography. The proposed methods were experimentally validated to be effective in segmenting plantar feet from thermal images under various conditions acquired by cold-stressed active thermography without complementary color or depth information, achieving an accuracy of 97.3% (Intersection over Union) and 95.4% (Dice Similarity Coefficient) on a specifically collected cold-stressed active thermography dataset, higher than that of several existing deep neural networks. Notably, PFSNet outperforms TransUNet, which incorporates a more sophisticated vision transformer into UNet, achieving better results with a smaller model and lower computational cost.

## Figures and Tables

**Figure 1 sensors-23-08511-f001:**
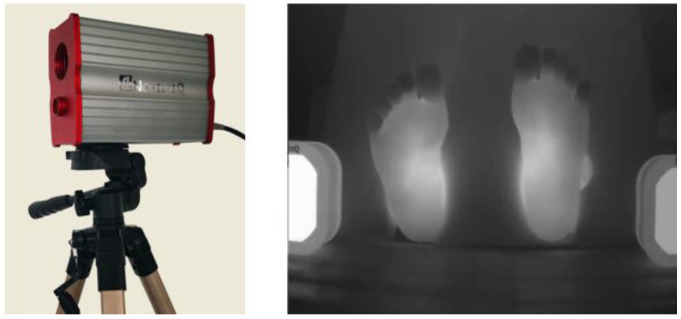
The NovelTEQ thermal camera and a thermogram with the two calibration bodies.

**Figure 2 sensors-23-08511-f002:**
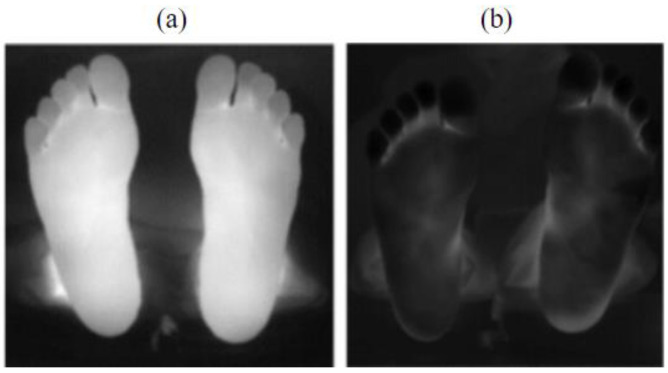
Thermal images of tested feet. (**a**) Plantar temperature surpassing the background temperature (high contrast image); (**b**) plantar temperature lower than or comparable to the background temperature (low contrast image).

**Figure 3 sensors-23-08511-f003:**
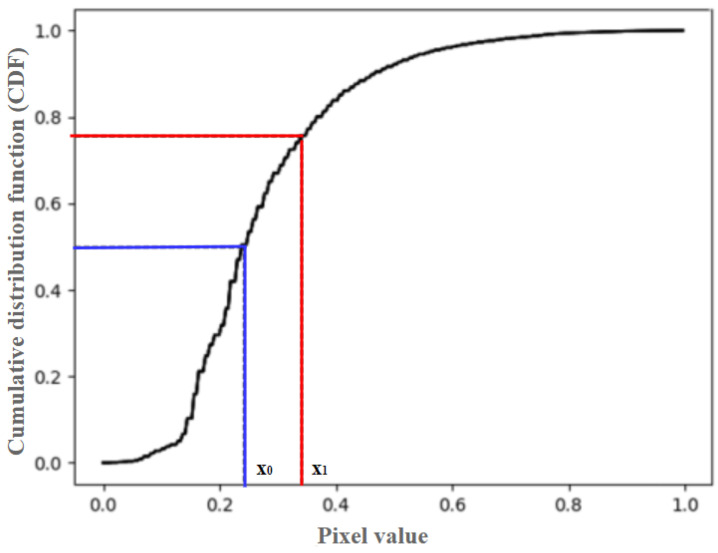
Cumulative distribution function of a thermal image.

**Figure 4 sensors-23-08511-f004:**
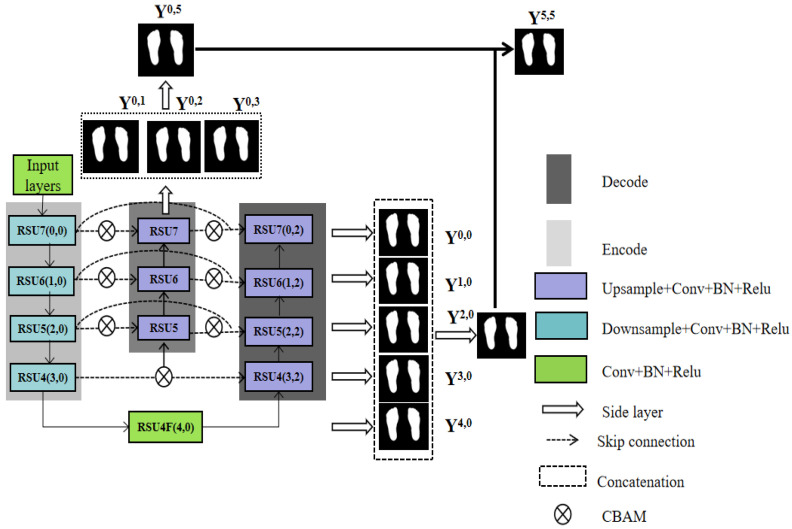
An overview of the proposed PFSNet architecture.

**Figure 5 sensors-23-08511-f005:**
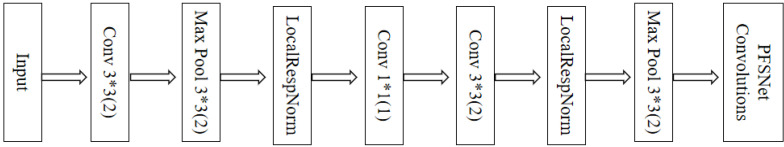
The initial input layers for PFSNet.

**Figure 6 sensors-23-08511-f006:**
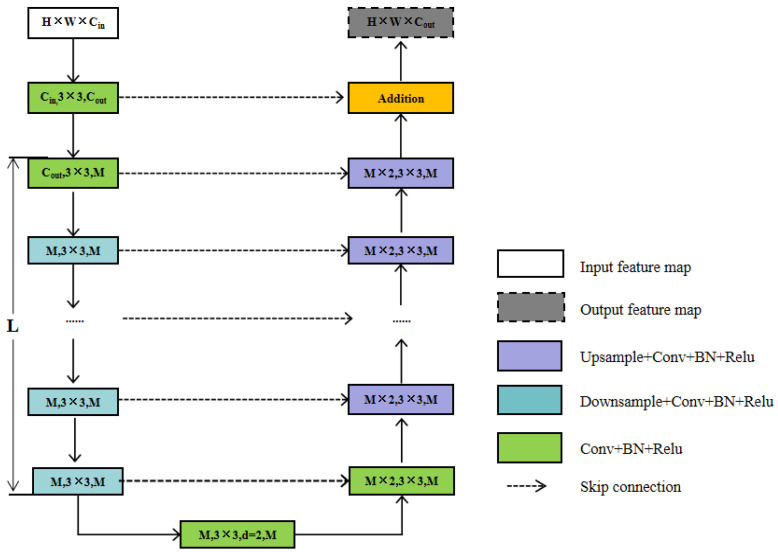
An overview of Residual U-block RSU-L.

**Figure 7 sensors-23-08511-f007:**
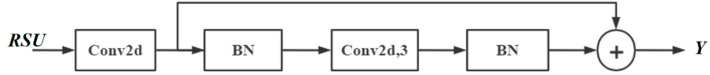
The fusion convolution unit.

**Figure 8 sensors-23-08511-f008:**
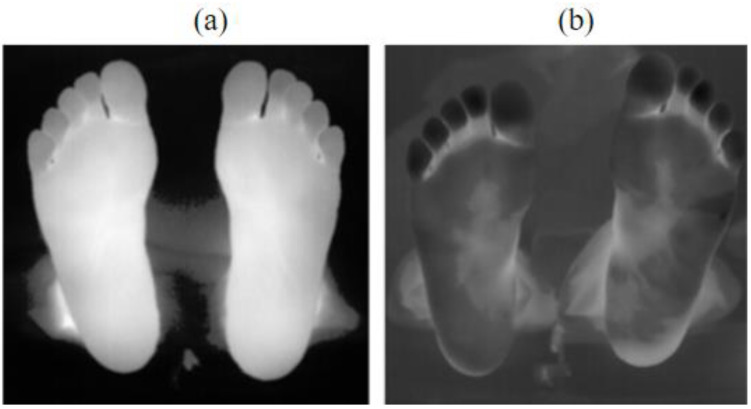
Results of the two-stage adaptive gamma transform for images in Figure 2. (**a**) High contrast image; (**b**) low contrast image.

**Figure 9 sensors-23-08511-f009:**
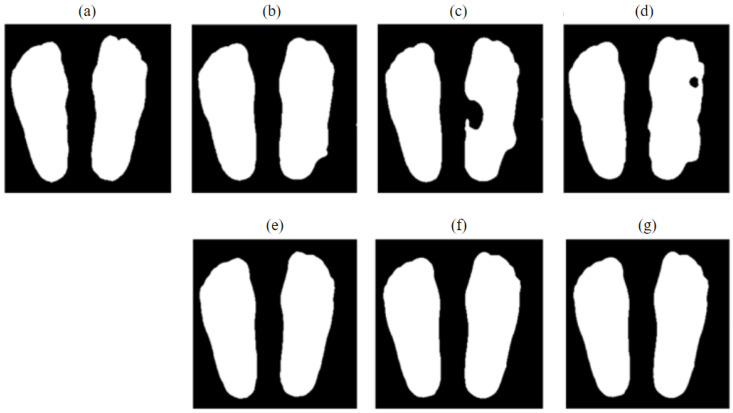
Segmentation results from different methods on plantar foot thermograms in Figure 2b. (**a**) Ground truth; (**b**) UNet; (**c**) UNet++; (**d**) AttentionUNet; (**e**) U2Net; (**f**) TransUNet; (**g**) PFSNet (proposed).

**Table 1 sensors-23-08511-t001:** Enhancement comparison of plantar foot thermal images using different methods. Bold number in each column represents the best result in each category.

Methods	Information Entropy	MSE	PSNR
Two-stage adaptive gamma transform	**5. 14**	**94. 36**	31. 58
Adaptive gamma transform	6. 15	99. 82	28. 33
Fixed gamma transform	5. 87	97. 86	27. 39
Histogram equalization	5. 96	98. 37	**32. 65**
No transform (original image)	6. 80	-	-

**Table 2 sensors-23-08511-t002:** The segmentation accuracy of PFSNet with and without image flipping and the proposed two-stage adaptive gamma transform. Bold number in each column represents the best result in each category.

Metrics/ Methods	HD (Mean ± STD)	DSC (Mean ± STD)	IOU (Mean ± STD)	Data Augmentation	
				Flipping	Proposed Transform
PFSNet	**0.962 ± 0.032**	**0.954 ± 0.005**	**0.973 ± 0.015**	Yes	Yes
PFSNet	0.954 ± 0.026	0.942 ± 0.016	0.961 ± 0.023	No	Yes
PFSNet	0.942 ± 0.043	0.938 ± 0.007	0.956 ± 0.035	No	No

**Table 3 sensors-23-08511-t003:** Segmentation accuracy of different methods on the plantar foot thermal image dataset. Bold number in each column represents the best result in each category.

Methods	HD (Mean ± STD)	DSC (Mean ± STD)	IOU (Mean ± STD)
PFSNet (proposed)	**0.962 ± 0.032**	**0.954 ± 0.005**	**0.973 ± 0.014**
TransUNet	0.941 ± 0.016	0.949 ± 0.011	0.951 ± 0.015
U^2^Net	0.952 ± 0.014	0.933 ± 0.021	0.943 ± 0.041
UNet ++	0.933 ± 0.042	0.931 ± 0.036	0.943 ± 0.043
AttentionUNet	0.918 ± 0.016	0.923 ± 0.018	0.916 ± 0.026
UNet	0.914 ± 0.037	0.853 ± 0.026	0.896 ± 0.032

## Data Availability

Data will be made available upon reasonable request.

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
