# Peer review of "Diabetic Plantar Foot Segmentation in Active Thermography Using a Two-Stage Adaptive Gamma Transform and a Deep Neural Network"

_sensors, 2023, doi:10.3390/s23208511_

Round 1

Reviewer 1 Report

Some of the techniques used in this work are essentially adaptations of existing methods and lack significant innovation. Furthermore, the experiments in this paper are inadequate, and the comparative models are outdated. The current experimental results fail to convince me of the value of the proposed method. Additionally, the writing logic in this paper is confusing. Therefore, I recommend that the editor rejects this manuscript. Below are some suggested revisions for reference.

1. In the section of introduction, the author provides an extensive list of previous works on deep neural networks for thermal image segmentation and highlights their success in this task. So, why did the author undertake this research? Did these methods have any shortcomings that inspired them to pursue this work? I believe this needs to be addressed to avoid the suspicion of contributing to an already saturated field.

2. To avoid ambiguity, do not use abbreviations in academic papers, for example, transform's.

3. In order to provide readers with a clearer understanding of your work, it is essential to supplement background information on thermal image segmentation and the foundational network architectures you employed.

4. Please briefly summarize the contributions of this work in the introduction.

5. There is a lack of an overall framework for the proposed method, making it difficult to directly grasp the relationships between different key components. It is recommended that the author supplements this and highlights your innovative contributions.

6. Regarding data augmentation, it appears that the author has only introduced what TSA is in the paper but hasn't specifically explained how data augmentation through TSA is implemented in this work concerning thermal images. Additionally, the author hasn't clearly explained what issues exist with thermal images that require data augmentation, and what results might be obtained if augmentation is not performed.

7. The deep neural network models used by the author for comparison appear to be quite outdated, dating back to at least 2020. It is recommended that the author includes comparisons with models proposed in the last two years.

8. It is advisable for the author to discuss the impact of data augmentation on the results in the experimental section.

9. The analysis of the experimental results is insufficient. The author has provided a simple description of the experimental results but hasn't conducted a thorough analysis, which is not comprehensive.

Extensive editing of English language required

Reviewer 2 Report

1.   The article lacks comprehensive details on the participant selection process, including specific criteria for inclusion and exclusion. This omission is critical as it affects the study's external validity and generalizability to the diabetic population. Did the article provide a detailed description of the participant selection process for the study? Were specific criteria for inclusion or exclusion of individuals in the research mentioned?

2.   The article does not specify the sample size used, nor does it mention whether a power analysis was conducted. This information is essential for assessing the statistical significance and reliability of the study's findings. What was the sample size used in the study, and was any power analysis conducted to determine if the sample size was adequate to detect expected differences?

3. The article indeed identifies gaps in the literature, specifically related to the application of thermography in diabetic foot care. However, the discussion could benefit from a more detailed emphasis on how this study addresses these gaps and contributes to the field. Have the authors identified any gaps in the existing literature that this study could fill, and is this contribution highlighted in the discussion?

4. The article does not delve into the potential long-term impacts of active thermography on diabetic foot complications, patient outcomes, or quality of life. This is a crucial aspect that requires further investigation. Can you provide information on the potential long-term impacts of using active thermography for diabetic foot complication detection, particularly in terms of patient outcomes and quality of life improvements?  

5. The article does not mention any technical limitations or drawbacks encountered during the system's implementation. This oversight leaves a gap in understanding the practical challenges associated with deploying thermography in clinical settings. Were there any technical limitations or drawbacks encountered during the implementation of the thermography-based system that were not discussed in the article?

6. The article lacks information on why the considerations for selecting thermal imaging equipment, such as resolution, sensitivity, and calibration. These factors are critical for the accuracy of thermal imaging. What considerations should be taken into account when selecting thermal imaging equipment for diabetic foot analysis that was not covered in the study?  

7. The article does not discuss any computational challenges or bottlenecks encountered during the development of the deep neural network. These issues can significantly impact the feasibility and efficiency of the proposed system. Were there any computational challenges or bottlenecks that arose during the development of the deep neural network for image segmentation that were not addressed in the article?

8. The article does not provide insights into unique or unconventional approaches to applying deep learning to thermal image analysis. This information could offer valuable insights into the innovation of the research. Can you discuss any unique or unconventional approaches used in the application of deep learning to thermal image analysis for diabetic foot complications that were not elaborated upon in the study?

9. The article does not mention any unanticipated integration challenges with existing healthcare IT systems. Such challenges are critical to understanding the successful integration of new technologies into clinical practice. Were there any unanticipated integration challenges with existing healthcare IT systems that emerged during the implementation of the thermography-based approach?

10. The article does not explore specific rehabilitation or therapeutic strategies enhanced by early detection through thermography. This is a missed opportunity to discuss potential clinical implications and benefits. In your opinion, what are the specific rehabilitation or therapeutic strategies that may be enhanced by early detection of diabetic foot complications through thermography, and why were these not covered in the article?

11. The article lacks elaboration on novel image enhancement or analysis techniques specific to thermal images. Such techniques could have a significant impact on the accuracy of diabetic foot complication detection. Can you elaborate on any novel image enhancement or analysis techniques specific to thermal images that were not detailed in the study?

12. The article does not touch upon regulatory or certification challenges in developing thermography-based systems. These are critical considerations for the translation of research into practical clinical applications. Were there any regulatory or certification challenges encountered in the development of thermography-based systems for diabetic foot care that were not discussed in the study?   

13. The article does not provide a comparative analysis of active thermography with other existing detection methods. Such a comparison is essential for assessing the relative effectiveness of thermography. How does the use of active thermography compare to other existing methods for early diabetic foot complications detection, such as Doppler ultrasound or clinical examination?

14. The article does not offer insights into the limitations or challenges of implementing the proposed system in a clinical setting. Understanding these challenges is crucial for practical application. Could you provide more information on the potential limitations or challenges of implementing the proposed thermography-based system in a clinical setting for diabetic patient care? 

15. Variations in equipment can significantly impact result reproducibility and validity. What specific thermal imaging equipment and hardware configurations were used in the study, and how might variations in equipment affect the accuracy and reliability of the results?

16. The article does not mention any plans to release code or pre-trained models, hindering the reproducibility and advancement of research in this area. Are there plans to release the code or pre-trained models used in this research to facilitate reproducibility and further experimentation in the field?   

17. The article does not discuss the integration of the system into healthcare IT infrastructure or potential challenges related to data management and interoperability. How do you envision the integration of the thermography-based system described in the article into existing healthcare IT infrastructure, and what are the potential challenges in terms of data management and interoperability?

18. The article does not delve into privacy and security concerns related to thermal image collection and storage, a critical aspect of healthcare technology development. Have you considered privacy and security concerns related to the collection and storage of thermal images from diabetic patients? How are these issues addressed in the proposed system?

19. The article does not mention plans for clinical trials or further validation studies, leaving uncertainty about the approach's clinical efficacy. Are there plans to conduct clinical trials or further studies to validate the effectiveness of the thermography-based approach in improving patient outcomes and reducing the incidence of severe foot complications?

20. The article does not address considerations of cost-effectiveness and scalability, which are essential for the real-world implementation of healthcare technologies. Have you considered potential cost-effectiveness and scalability issues when deploying thermography-based systems in healthcare settings with a high volume of diabetic patients?

21. Have you considered potential cost-effectiveness and scalability issues when deploying thermography-based systems in healthcare settings with a high volume of diabetic patients?

"The data were analyze." > "The data were analyzed."

"The model has ran successfully." > "The model has run successfully."

"The Harvard University conducted the research." > "Harvard University conducted the research."

"Researchers must to conduct further studies." > "Researchers must conduct further studies."

"A new softwares were developed." > "New software was developed."

"The results were obtained in the lab." > "The results were obtained at the lab."

"The patient's symptoms was concerning." > "The patient's symptoms were concerning."

"Patients with the diabetes are at risk." > "Patients with diabetes are at risk."

"The study was very detailed." > "The study was highly detailed."

Round 2

Reviewer 1 Report

This paper proposed a novel deep neural network model, PFSNet, that employs a U-shaped network and multi-scale feature extraction to localize foot boundaries in plantar foot thermal images.

However, the paper innovation points are not clearly expressed and has an average writing style. And this manuscript suffers from a number of weak points, it should be further improved before consideration for publication. Let's elaborate on some of them:

1)Please revise the abstract section to condense the context of the research. Clearly express the purpose and contribution of the study and highlight the innovation of the paper.

2)I suggest the authors make a comprehensive investigation of the diabetic foot detection methods in the literature in the introduction part and give the analysis to the existing works such as (https://doi.org/10.1016/j.compbiomed.2021.104596,https://doi.org/10.1016/j.compbiomed.2021.104838,https://doi.org/10.1016/j.compbiomed.2021.104491) to make the whole work more in-depth.

3)Section 2.4 appears twice in the paper chapter. Authors are requested to double-check and make corrections.

4)Please pay attention to the formatting of the conclusion section, and please be consistent with the formatting of the other sections by enforcing first line indentation.

5)Expand the critical results in the conclusion. Focus on the main developments in the finale. Also, write the main contributions in the conclusion.

Minor editing of English language required.
